# Routine Tracheal Intubation and Meconium Suctioning in Non-Vigorous Neonates with Meconium-Stained Amniotic Fluid: A Systematic Review and Meta-Analysis

**DOI:** 10.3390/diagnostics12040881

**Published:** 2022-04-01

**Authors:** Maria Dikou, Theodoros Xanthos, Ioannis Dimitropoulos, Zoi Iliodromiti, Rozeta Sokou, Georgios Kafalidis, Theodora Boutsikou, Nicoletta Iacovidou

**Affiliations:** 1Department of Neonatology, Aretaieion University Hospital, Medical School, National and Kapodistrian University of Athens, 115 28 Athens, Greece; mariadikou@gmail.com (M.D.); i.dimitropoulos@nhs.net (I.D.); ziliodromiti@yahoo.gr (Z.I.); sokourozeta@yahoo.gr (R.S.); gkafalidis@gmail.com (G.K.); theobtsk@gmail.com (T.B.); 2Medical School, European University Cyprus, Nicosia 2404, Cyprus; theodorosxanthos@yahoo.com

**Keywords:** endotracheal intubation, endotracheal suction, meconium-stained amniotic fluid, meconium-stained infant, non-vigorous infant

## Abstract

The aim of this systematic review and meta-analysis is the comparison of endotracheal intubation and suctioning to immediate resuscitation without intubation of non-vigorous infants > 34 weeks’ gestation delivered through meconium-stained amniotic fluid (MSAF). Randomized, non-randomized clinical trials and observational studies were included. Data sources were PubMed/Medline and Cochrane Central Registry of Controlled Trials, from 2012 to 2021. Inclusion criteria were non-vigorous infants born through MSAF with gestational age > 34 weeks and sample size ≥ 5. We calculated overall relative risks (RR) and mean differences (MD) with a 95% confidence interval (CI) to determine the impact of endotracheal suction (ETS) in non-vigorous infants born through MSAF. The outcomes presented are the incidence of neonatal mortality, meconium aspiration syndrome (MAS), transient tachypnea, need for positive pressure ventilation, respiratory support, persistent pulmonary hypertension treatment, neonatal infection, ischemic encephalopathy, admission to neonatal intensive care unit (NICU) and the duration of hospitalization between ETS and non-ETS group. Six studies with a total sample of 1026 patients fulfilled the inclusion criteria. Statistically non-significant difference was observed in RR between two groups with regards to mortality (1.22, 95% CI 0.73–2.04), occurrence of MAS (1.08, 95% CI 0.76–1.53) and other outcomes, and MD in hospitalization duration. There is no sufficient evidence to suggest initiating endotracheal suction soon after birth in non-vigorous meconium-stained infants as routine.

## 1. Introduction

The presence of meconium in the amniotic fluid following rupture of the membranes during childbirth is a complication that affects approximately 13% of all births (7–20%) [1] and characterizes the amniotic fluid as stained. It is a sign of intrauterine hypoxia [2]. Conditions that may lead to the release of amniotic fluid include hypoxia, acidosis, placental insufficiency, oligohydramnios, umbilical cord compression, infection, and hypertension-preeclampsia. About 20% of newborns born with stained amniotic fluid are non-vigorous and are more likely to experience respiratory distress and its effects, while 4–5% of newborns treated in neonatal intensive care units (NICU) manifest meconium aspiration syndrome (MAS) [3]. MAS is defined as respiratory distress that occurs in neonates with stained amniotic fluid, which cannot be attributed to other causes. It is the most feared complication of the aspiration of stained amniotic fluid, affecting about 3–9% of cases of stained amniotic fluid (1–2/1000 live newborns). It is initially characterized by chemical pneumonia and can be complicated by air leakage syndromes (pneumothorax, subcutaneous emphysema, intermediate pulmonary emphysema) and persistent pulmonary hypertension [4]. Therefore, the presence of meconium-stained amniotic fluid indicates an increased probability of the necessity for resuscitation of the newborn at birth [5].

Since the 1970s and 1980s, interventions have been proposed and implemented to reduce neonatal mortality and morbidity. Based on the results of a large randomized clinical trial that occurred in 1976, oropharyngeal aspiration was performed upon delivery of the baby’s head at the perineum, followed by endotracheal intubation and suction of the trachea at birth, as a regular practice for several of decades [1]. This practice was abandoned about 15 years later, when it was shown that vigorous neonates with meconium-stained amniotic fluid did not benefit from the invasive endotracheal suction and intubation; as per the American Academy of Pediatrics (AAP) and the International Resuscitation Committee guidance, endotracheal intubation and suction were then only performed for hypotonic, weakly crying neonates [5,6].

Since 2015, the American Academy of Pediatrics, the American College of Gynecologists (ACOG), the American Heart Association (AHA), and the International Resuscitation Committee (International Liaison Committee on Resuscitation) have recommended the vigorous newborn be brought next to the mother, while for the hypotonic, non-crying or bradycardic neonate, the endotracheal suction to be abandoned as a routine intervention and to be substituted by clearing of the upper airways from meconium under direct visibility [6]. In addition, lung expansion and ventilation are recommended as the primary measures for immediate oxygenation, while suction can be performed when the airway is blocked by meconium [7]. However, it remains controversial whether the method of endotracheal intubation and suction in hypotonic, non-crying, or bradycardic neonates positively affects the outcome [8].

There are insufficient published data to suggest routine tracheal intubation for meconium suction in non-vigorous neonates as opposed to non-endotracheal intubation and suction. The most recently published clinical studies after 2015 showed that endotracheal suction does not affect the outcome in these neonates, including mortality and MAS [9,10].

## 2. Materials and Methods

This is a systematic review and meta-analysis of randomized and non-randomized clinical trials, before-and-after studies, and cohort studies that determine the effectiveness of no intervention with endotracheal suction compared to routine endotracheal suction in non-vigorous neonates born with meconium-stained amniotic fluid and the presentation of the outcomes.

Original systematic literature search: A systematic literature search was conducted to identify relevant studies published from 2012 until February 2021 in the following databases: Medline (PubMed) and Cochrane Central Registry of Controlled Trials. The search was conducted using two groups of keywords proposed by a group of experts with relevant methodological and clinical expertise. The Medical Subject Headings (MeSH) database was used for the identification of synonyms. These two groups were combined by the Boolean “AND” and the terms used within these search categories were combined by the Boolean “OR”. The full search strategy used for PubMed was: Endotracheal suction OR Endotracheal intubation OR intubation (title/abstract) AND Meconium-stained amniotic fluid OR Meconium stained infant OR Meconium (title/abstract). This was adapted appropriately for the rest of the databases. Reference lists of selected articles were used to find additional studies that were not retrieved in the initial search. Conference abstracts were not searched because they do not contain enough data for quality assessment.

Study selection process: The study selection process is documented in the respective flow chart (Figure 1). Following the original systematic literature search, duplicates were excluded. The remaining studies were evaluated by two independent researchers according to pre-determined inclusion and exclusion criteria, which were determined based on the PICOS (population, intervention, comparator, outcomes, setting) question format.

Randomized (RCT) and non-randomized (non-RCT) clinical trials, before-and-after studies, and cohort studies were eligible for inclusion, whereas case reports, case-control studies, review studies (systematic reviews and meta-analyses), and letters to the editor were excluded. Studies should include non-vigorous neonates born after 34 weeks of gestation; studies with non-human populations and small case series (<5 patients) were excluded. Only articles written in English were eligible. Study authors were contacted, when appropriate, to request additional unpublished data.

The study selection was performed using a two-step process. In the first step, researchers assessed the studies based on their title and abstract. For the articles where the data provided in the title or abstract were insufficient to decide on inclusion, the full text was assessed against the inclusion and exclusion criteria (second step). Studies with no available full text were excluded from the systematic review and meta-analysis.

Data extraction: The same two independent researchers conducted the data extraction using a standardized data extraction form developed to serve the purposes of the study. Potential discrepancies between the two researchers were resolved by consensus among them or by a third investigator. The following data were extracted: general characteristics of the study (first author, publication year, country), study design, and sample characteristics (sample size, sex distribution, gestational age, birth weight, neonatal situation, and maternal parameters). The outcomes extracted were neonatal death, occurrence of meconium aspiration syndrome, transient tachypnea, need for respiratory support or mechanical ventilation, treatment for persistent pulmonary hypertension, ischemic encephalopathy, infection, admission to NICU, and duration of hospital stay.

Data synthesis and analysis: This study was conducted in accordance with the recommendations of the Preferred Reporting Items for Systematic Reviews and Meta-analyses (PRISMA) statement. The study protocol included the following steps: (a) original systematic literature search, (b) selection of appropriate studies to be included following application of the inclusion and exclusion criteria, (c) data extraction, and (d) data synthesis and analysis.

The effect size used was the relative risk (RR) of the occurrence of the outcomes between the two groups of neonates: endotracheal suction group (ETS) and non-endotracheal suction group (non-ETS), and the mean difference (MD) for the outcome of hospitalization duration.

The meta-analysis was conducted using the STATA command “metan” to estimate the pooled effect size with a 95% confidence interval (CI). The results are presented in the form of a forest plot. The pooled effect size was estimated through pooled RR and pooled MD and was considered statistically significant when the 95% CI did not include the 1 value and the 0 value for the two sizes, respectively.

The I^2^ statistic was used to assess the heterogeneity among included studies. When the *p*-value of the statistic was <0.05, the heterogeneity was considered statistically significant, and in such a case, the random-effects model (REM) using the method of Der Simonian and Laird (D + L) was used to obtain the pooled effect sizes. When *p* > 0.05, the heterogeneity was considered statistically non-significant, and thus a fixed effect model was used. A general interpretation of the I^2^ value is as follows: 0% to 40% might not be important heterogeneity, 30% to 60% may represent moderate heterogeneity, 50% to 90% substantial heterogeneity, and 75% to 100% considerable heterogeneity.

The publication bias was assessed using the funnel plot and the statistical test of Egger. As for Egger’s test, *p* < 0.05 indicates statistically significant publication bias. All analyses were performed with the STATA v.15 software.

## 3. Results

The stepwise study selection process is summarized in the flow chart of the systematic review (Figure 1). Following exclusion of duplicate studies, 7563 studies were initially identified. Of those, 7471 studies were excluded during the process of assessing the title and/ or the abstract. The remaining 92 studies were further evaluated by assessing the full text, and 6 of them fulfilled all the pre-determined inclusion and exclusion criteria to be included in the current systematic review and meta-analysis.

Study characteristics: The characteristics of the six studies included in the meta-analysis were as follows: four studies were conducted in India, one in the USA, and one in the United Kingdom. All were published between 2015 and 2020 [11,12,13,14,15,16] (Table 1). Four studies were randomized clinical trials (RCT) [11,12,13,14], and two were observational studies [15,16].

The neonates included in the studies were non-vigorous and were defined as follows: heart rate lower than 100 beats per minute, decreased muscle tone, gasping, or not breathing/ crying. All were delivered through meconium-stained amniotic fluid. The intervention evaluated was endotracheal suction versus non-endotracheal suction in the above neonates. The four RCT studies, namely Nangia et al. [13], Kumar et al. [12], Chettri et al. [11], and Singh et al. [14], in order to evaluate the effect of non-endotracheal suction on the occurrence of MAS and all-cause mortality in non-vigorous neonates born through MSAF, randomized 581 non-vigorous neonates born through MSAF with GA > 34 weeks, in two groups, the endotracheal suction (ETS) group, and the non-endotracheal suction (non-ETS) group. All studies compared endotracheal intubation and suctioning with no endotracheal intubation and suctioning. Observational studies of Chiruvolu et al. [15] and Oommen et al. [16] included 231 and 229 non-vigorous neonates, respectively, born through MSAF with GA > 35 weeks and compared the retrospective 1-year period (2015–2016) and prospective 1-year period (2016–2017) as far as the 2015 Neonatal Resuscitation Programme (NRP) guidelines are concerned. Five studies set survival as the predominant outcome [11,12,13,14,15], while all six the possibility for MAS [11,12,13,14,15,16].

Five studies provided data on the need for respiratory support or mechanical ventilation [12,13,14,15,16]. Four studies provided results for the need for PPV therapy [11,12,13,14]. Two studies provided results for the risk of TTN [12,13,14,15,16], and four studies provided results for the need for treatment of PPHN [11,12,14,15]. In addition, five studies provided results for the risk of ischemic encephalopathy [11,12,13,14,15] and three studies for the risk of infection [11,12,14]. Finally, regarding the possibility of admission of these newborns to the NICU, two of the studies yielded results [15,16], while three studies provided results regarding the total duration of hospitalization [13,14,15].

Patients’ characteristics: Table 1 shows the patient characteristics of included studies. In the RCT studies, patient characteristics were well-matched between the groups. In the observational studies, incidences of late-preterm and post-term infants were lower in the intervention group than in the comparator group, although the mean GA was similar between the two groups with a mean value (standard deviation) of 39.9 (1.5) vs. 39.9 (1.1) weeks. Fetal distress was significantly lower in the intervention group.

Outcome analysis: All studies compared laryngoscopy with endotracheal intubation and suctioning with immediate resuscitation without laryngoscopy. The outcomes of the studies are presented in Table 2.

### 3.1. Death

The outcome of death was reported in five studies [11,12,13,14,15], and one of them had no deaths [15]. Statistically non-significant heterogeneity was detected between studies (I^2^ = 17.1%, *p* = 0.306), so a fixed-effects model was used. This analysis revealed that neonates who underwent ETS had a 22% higher risk of dying compared to infants who did not undergo ETS. However, this difference was statistically non-significant (RR 1.22, 95% CI 0.73–2.0, Figure 2a). In addition, the funnel plot and Egger’s test (*p* = 0.827) did not show significant publication bias.

### 3.2. Meconium Aspiration Syndrome (MAS)

Six studies included data on the cumulative RR for the occurrence of MAS between ETS and non-ETS [11,12,13,14,15,16]. Due to the statistically significant heterogeneity between the studies (I^2^ = 65.8%, *p* = 0.012), a random-effects model was used. Infants who underwent ETS presented with an 8% higher risk of developing MAS compared to infants who did not undergo ETS. However, this difference was not statistically significant (RR 1.08, 95% CI 0.76–1.53, *p* = 0.684, Figure 2b). In addition, both funnel plot and Egger’s test (*p* = 0.817) did not reveal significant publication bias.

### 3.3. Respiratory Support or Mechanical Ventilation

Five studies estimated the cumulative RR for respiratory support between ETS and non-ETS [12,13,14,15,16]. No statistically significant heterogeneity was found between the studies (I^2^ = 50.5%, *p* = 0.089), so a fixed-effects model was used. Infants who underwent ETS had a 4% lower risk of respiratory support than infants who did not undergo ETS; this difference was not statistically significant (RR 0.96, 95% CI 0.84–1.10, *p* = 0.573, Figure 3a). Funnel plot and Egger’s test (*p* = 0.817) did not show significant publication bias.

### 3.4. Transient Tachypnea of Newborn

Cumulative RR for TTN between ETS and non-ETS was included in two studies [12,16], which did not present statistically significant heterogeneity (I^2^ = 60.4%, *p* = 0.112); thus, a fixed-effects model was used. Analysis revealed that compared to those that did not undergo ETS, infants that ETS was performed had a 27% higher risk of TTN. However, this difference was not statistically significant (RR 1.27, 95% CI 0.88–1.84, *p* = 0.199, Figure 3b). Due to the small number of studies, the publication bias could not be evaluated.

### 3.5. Positive Pressure Ventilation

Four studies reported data on the cumulative RR for PPV [11,12,13,14]. Due to statistically significant heterogeneity between studies (I^2^ = 62.2%, *p* = 0.047), a random-effects model was used. This analysis revealed similar risk among the two groups (RR 1.00, 95% CI 0.87–1.14, *p* = 0.983, Figure 3c). In addition, funnel plot and Egger’s statistical test (*p* = 0.380) did not show significant publication bias.

### 3.6. Persistent Pulmonary Hypertension Treatment

Four studies estimated the cumulative RR for treatment of PPH between ETS and non-ETS [11,12,14,15] with statistically non-significant heterogeneity (I^2^ = 6.9%, *p* = 0.359), so a fixed-effects model was used. Analysis revealed a 5% lower risk for infants who underwent ETS, which was not statistically significant (RR 0.95, 95% CI 0.50–1.80, *p* = 0.865, Figure 3d). Funnel plot and Egger’s test (*p* = 0.300) did not show significant publication bias.

### 3.7. Ischemic Encephalopathy

Five studies estimated the cumulative RR for ischemic encephalopathy [11,15]. No statistically significant heterogeneity was observed (I^2^ = 0.0%, *p* = 0.598), thus a fixed-effects model was used. This analysis revealed that infants who underwent ETS had a 12% lower risk of developing ischemic encephalopathy than the non-interventional group. However, this difference was not statistically significant (RR 0.88, 95% CI 0.68–1.14, *p* = 0.349, Figure 4a). Funnel plot and Egger’s test (*p* = 0.612) did not show significant publication bias.

### 3.8. Infection

Three studies estimated the cumulative RR for the occurrence of infection between the two groups [11,12,14]. No statistically significant heterogeneity was found between the studies (I^2^ = 21.2%, *p* = 0.281), so a fixed-effects model was used. Infants who underwent ETS had a 32% higher risk; however, this difference was not statistically significant (RR 1.32, 95% CI 0.48–3.57, *p* = 0.590, Figure 4b). Due to the small number of studies, the publication bias could not be evaluated.

### 3.9. Admission to NICU

Two studies estimated the cumulative RR for neonatal admission to NICU [15,16]. Due to statistically significant heterogeneity between studies (I^2^ = 97%, *p* < 0.001), a random-effects model was used. Oommen et al. showed a reduction in admission to NICU of non-ETS newborns (*p* < 0.05) [16]. However, the pooled analysis revealed that infants who underwent ETS had a 28% higher risk of being admitted to NICU. Pooled ratio was not statistically significant (RR 1.28, 95% CI 0.26–6.43, *p* = 0.763, Figure 5a). Due to the small number of studies, the publication error could not be evaluated.

### 3.10. Duration of Hospital Stay

Finally, three studies estimated the MD difference in hospitalization duration [13,14,15]. Due to statistically significant heterogeneity between studies (I^2^ = 68%, *p* = 0.044), a random-effects model was used. Results revealed that infants who underwent ETS had 0.62 fewer days of hospitalization than infants who did not undergo ETS; this difference was not statistically significant (MD −0.62, 95% CI −1.72–0.47, *p* = 0.262, Figure 5b). Due to the small number of studies, the publication bias could not be evaluated.

## 4. Discussion

The 2015 and the most recent 2021 guidelines of the American Academy of Pediatrics (AAP), the American College of Obstetricians and Gynecologists (ACOG), the American Heart Association (AHA), and the International Liaison Committee on Resuscitation (ILCOR) declare there is insufficient published scientific evidence to suggest endotracheal suction and intubation as a routine method for non-vigorous neonates born with meconium-stained amniotic fluid compared to non-endotracheal suction. Moreover, suction in hypotonic, non-crying neonates may lead to a delay in initial efficient ventilation, although some neonates with meconium airway obstruction need aspiration or intubation for subsequent ventilation [17].

Due to insufficient data and the fact that the above recommendation is not based on well-designed clinical trials, this controversial issue was a major topic for discussion in the International Recovery Committee both in 2015 and 2021 [9,10]. Since 2015, new studies have been published on the topic, which are all included in this systematic review and meta-analysis. Our systematic review and meta-analysis included 6 studies with a total sample of 1026 hypotonic, non-crying, or bradycardic neonates born with MSAF. Results revealed that the resuscitation approach based on direct laryngoscopy and endotracheal aspiration did not show worse outcomes in terms of infant mortality compared to immediate resuscitation without laryngoscopy and endotracheal intubation.

Similar results were obtained by comparing the two methods regarding the likelihood of MAS and other respiratory complications, such as TTN. Moreover, neonates who underwent endotracheal suction were at the same risk for respiratory support with mechanical ventilation or positive airway pressure application compared to those that did not undergo endotracheal suction.

Similarly, statistically non-significant differences emerged between the two groups with regard to the need for treatment for PPH. Other parameters studied, such as the likelihood of developing ischemic encephalopathy and infection, were found with comparable results with statistically non-significant differences. Lastly, results were also similar regarding admission of newborns to NICU as well as for the duration of their hospitalization.

Although Chiruvolu et al. revealed an increase in NICU admission due to respiratory problems and a need for therapy in non-vigorous, meconium-stained infants [15] after the 2015 NRP guidelines and Singh et al. revealed an increase in the duration of hospitalization of the non-ETS group [14], results in overall yielded no difference in the outcomes of neonates receiving or not receiving ETS.

The results of the present study are consistent with those of the systematic review and meta-analysis of Trevisanuto et al. in 2020 [18], which included four clinical trials and an observational study, and resulted in a statistically non-significant difference between the intubated and suctioned and the non-intubated group of hypotonic, non-crying group neonates with stained amniotic fluid in terms of survival and the occurrence of MAS and hypoxic ischemic encephalopathy. Similar were the results of the systematic review and meta-analysis of Phattraprayoon et al. in 2020 [19], which included four clinical trials, and of Nangia et al. in 2021 [20], which also included the same four clinical trials.

### 4.1. Limitations

There are several limitations in this study, primarily the fact that it consists of heterogeneous studies, such as clinical trials and observational studies, which have different quality and design. In this systematic review and meta-analysis, four randomized clinical trials and two observational studies were included. Particularly regarding four outcomes, namely the occurrence of MAS, respiratory support via the use of PPV, the likelihood of admission to NICU, and the duration of hospitalization, statistically significant heterogeneity was observed among the studies. Moreover, only two studies [12,13] estimated the cumulative RR for TTN and two studies [15,16] for neonatal admission to NICU between ETS and non-ETS groups. The observational studies were considered for review in our meta-analysis as our protocol included before-and-after and cohort studies; the inclusion of the observational studies allowed the increase in the sample size. The RCTs were conducted in a low-resource setting, while the two observational were conducted in a high-resource one. There is a likelihood that the results would have been different if the RCTs had been conducted in high-resource settings. It must be noted that we did not perform a de novo quality appraisal of the primary studies as it has already been performed in previous systematic reviews published [18,20].

A possible limitation of our review is to address the 2015 change in guidelines; thus, we include studies from three years before the recommendation until now.

None of the included studies reports the number of attempts made for successful intubation or the time required for successful ventilation with positive pressure [21]. In addition, there is no precise reference to the meconium volume aspirated from the neonatal airways, although large amounts of meconium appear to have been aspirated from the neonatal airways with MAS [22]. Meconium-stained amniotic fluid encompasses rather heterogeneous conditions related mainly (but not only) to the density of the fluid itself with different effects on the respiratory system. Any attempt to pool these conditions together would fail in the definition of the optimal treatment of each specific condition itself. Therefore, the present findings do not constitute a strong recommendation.

Both the findings and limitations of our systematic review and the shortcomings of the literature suggest that the appropriate intervention in the recovery of hypotonic neonates delivered through MSAF should be further explored.

### 4.2. Clinical Application

In conclusion, the present study does not demonstrate a clear superiority of one of the two methods of intervention, namely direct endotracheal intubation and suction versus non-invasive ventilation without endotracheal suction. For hypotonic, non-crying neonates, the data so far are not sufficient to suggest immediate endotracheal suction as a routine intervention. The most recent international guidelines indicate intubation and aspiration for thick meconium-stained amniotic fluid producing an obstructive plug in the trachea or bronchi. At this point, it should be mentioned that the risks or benefits of intubation with tracheal suctioning may vary depending on the gestational age, the thickness of meconium, or the operator’s experience. Therefore, further studies are required that meet high-quality criteria with sufficient sample size and appropriate methodology. We are not able at this stage to suggest a specific number of cases, as a specific power analysis was not in the scope of our study; however, it is something that will concern us in the near future. We suggest that more randomized, controlled, double-blind clinical trials be conducted, including the consideration of factors such as the thickness of the stained amniotic fluid, the GA and the operator’s experience, and the starting point of the intervention. On the other hand, such studies involve ethical concerns and difficulty in implementation (voluntary withdrawal of patients from the intervention group). It is, however, essential to accurately investigate both the degree of effectiveness of the intervention and the optimization of the skills of the resuscitators who apply the respective intervention.

## Figures and Tables

**Figure 1 diagnostics-12-00881-f001:**
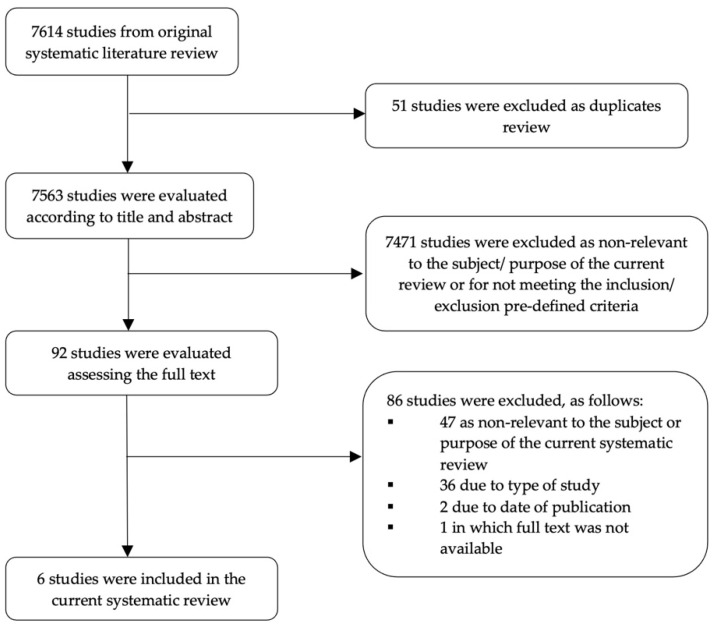
Flow chart of the stepwise study selection process.

**Figure 2 diagnostics-12-00881-f002:**
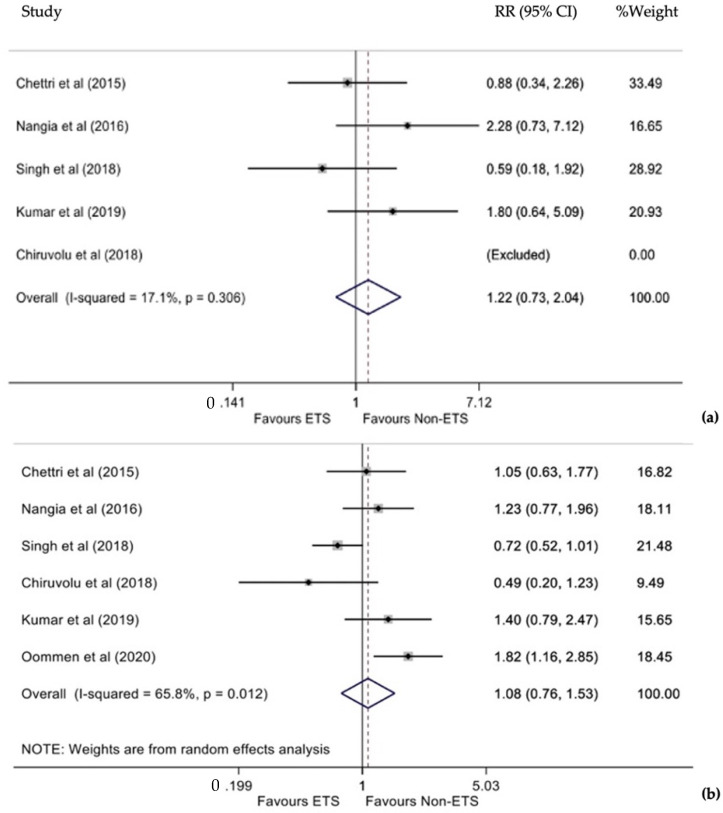
(**a**) Forest plot of relative risk (RR) for death between endotracheal suction (ETS) and non-endotracheal suction (non-ETS). (**b**) Forest plot of RR for MAS between ETS and non-ETS.

**Figure 3 diagnostics-12-00881-f003:**
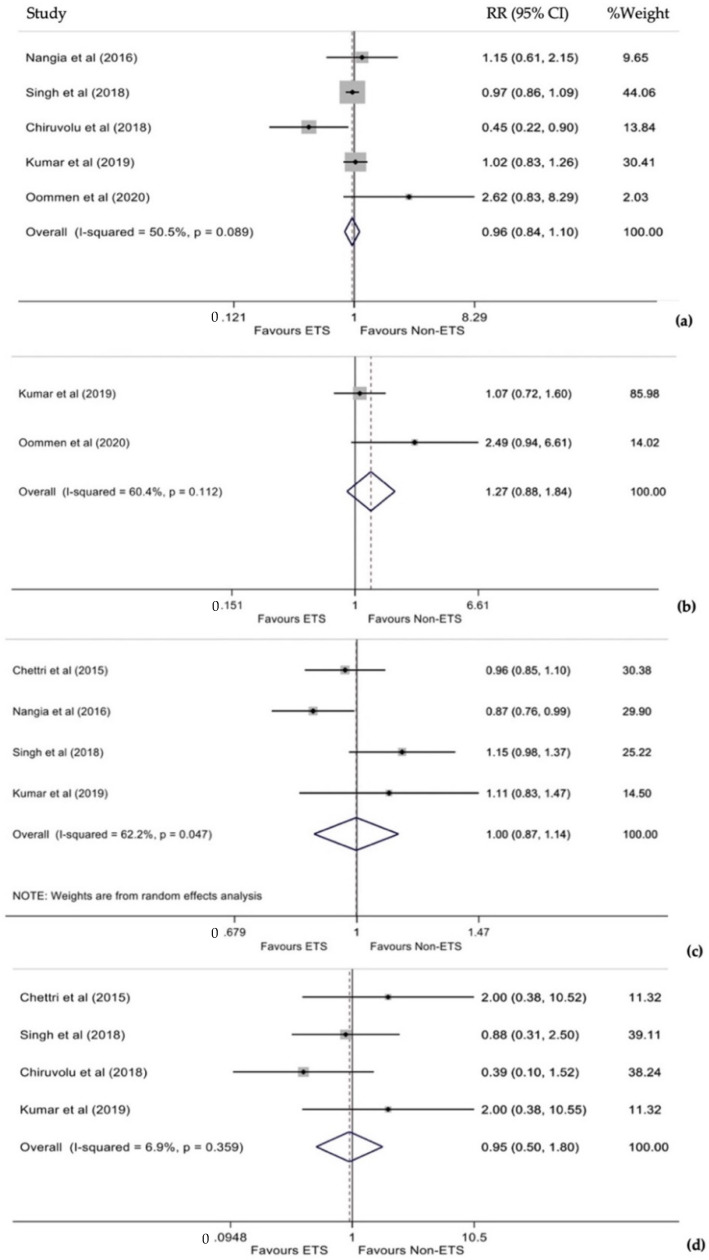
(**a**) Forest plot of RR for respiratory support or mechanical ventilation between ETS and non-ETS. (**b**) Forest plot of RR for TTN between ETS and non-ETS. (**c**) Forest plot of RR for PPV between ETS and non-ETS. (**d**) Forest plot of RR for PPHN between ETS and non-ETS.

**Figure 4 diagnostics-12-00881-f004:**
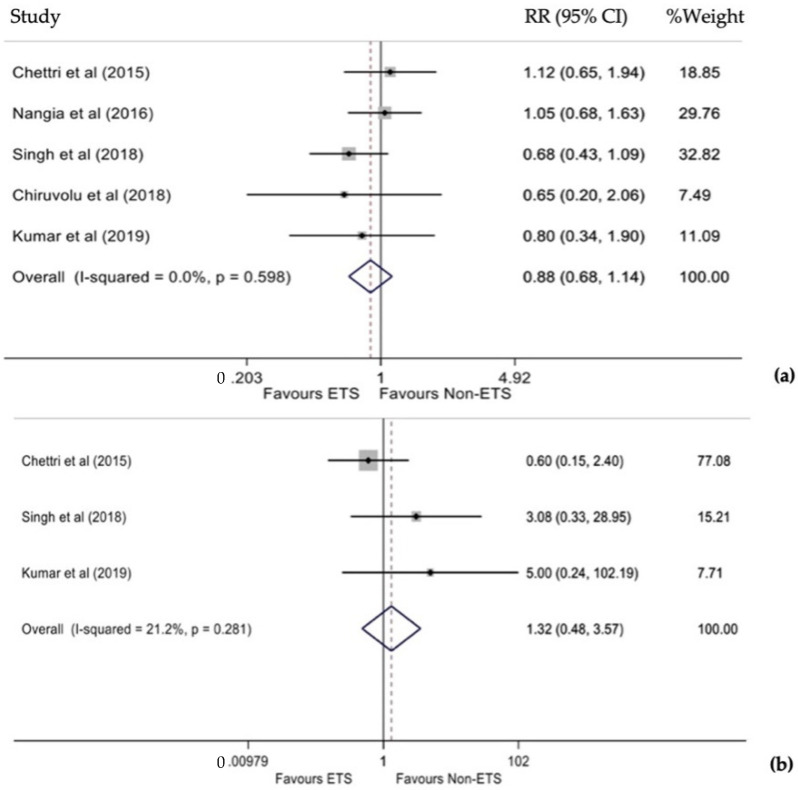
(**a**) Forest plot of RR for ischemic encephalopathy ETS and non-ETS. (**b**) Forest plot of RR for infection between ETS and non-ETS.

**Figure 5 diagnostics-12-00881-f005:**
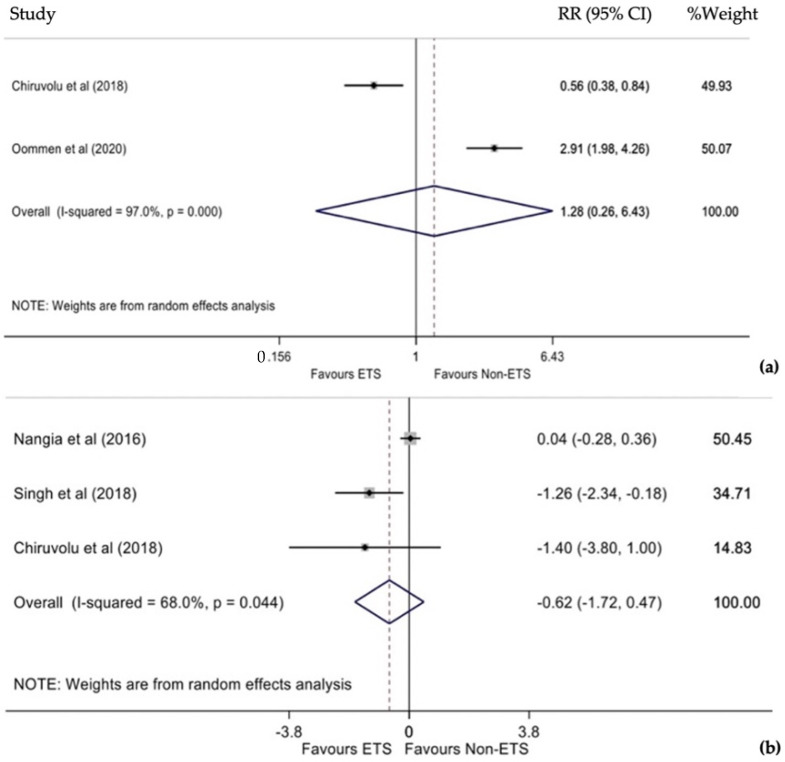
(**a**) Forest plot of RR for admission to NICU between ETS and non-ETS. (**b**) Forest plot of mean difference in duration of hospitalization ETS and non-ETS.

**Table 1 diagnostics-12-00881-t001:** Studies and patients’ characteristics per study (in ascending chronological order).

Study, Year	Country	Study Design	Intervention	Number of Neonates (Boys)	GA (Weeks), Mean ± SD	BW (Kg), Mean ± SD	Number of Neonates with Apgar Score < 7 at 5th min
Chettri et al., 2015	India	RCT	ETS	61 (35)	37–42	2.87 ± 0.49	30
			Non-ETS	61 (36)		2.90 ± 0.35	27
Nangia et al., 2016	India	RCT	ETS	87 (52)	39 (37–40)	2.649 ± 0.437	16
			Non-ETS	88 (52)		2.763 ± 0.533	13
Singh et al., 2018	India	RCT	ETS	75 (40)	38.5 ± 2.0	2.462 ± 0.315	30
			Non-ETS	77 (43)		2.461 ± 0.192	26
Kumar et al., 2019	India	RCT	ETS	66 (35)	38 (36–40)	2.620 ± 0.696	13
			Non-ETS	66 (29)		2.528 ± 0.598	15
Chiruvolu et al., 2018	USA	OBS	ETS	130 (64)	39.9 ± 1.0	2.453 ± 0.549	23
			Non-ETS	101 (62)		3.397 ± 0.620	22
Oommen et al., 2020	U.K.	OBS	ETS	68	39.6 ± 1.3	3571.9 ± 0.452.3	n/a
			Non-ETS	146	40.2 ± 1.2	3657.1 ± 0.468.6	n/a

GA: gestational age; BW: birth weight; ETS: endotracheal suction; Non-ETS: non-endotracheal suction; RCT: randomized controlled trial; OBS: observational study; n/a: data not available.

**Table 2 diagnostics-12-00881-t002:** Summary of the outcomes of studies (in ascending chronological order) between the ETS and non-ETS groups.

	Study, Year	Chesttri et al., 2015		Nangia et al., 2016		Singh et al., 2018		Kumar et al., 2019		Chirovolu et al., 2018		Oomen et al., 2020	
Outcome		ETS	Non-ETS	ETS	Non-ETS	ETS	Νon-ETS	ETS	Non-ETS	ETS	Non-ETS	ETS	Non-ETS
Death		7	8	9	4	4	7	9	5	0	0	n/a	n/a
MAS		20	19	28	23	31	44	21	15	7	11	25	30
Transient tachypnea		n/a	n/a	n/a	n/a	n/a	n/a	29	27	n/a	n/a	n/a	n/a
Respiratory support		n/a	n/a	17	15	65	69	48	47	11	19	6	5
Positive pressure ventilation		53	55	68	79	53	56	41	37	n/a	n/a	n/a	n/a
Treatment for pulmonary hypertension		4	2	n/a	n/a	6	7	4	2	3	6	8	7
Ischemic encephalopathy		19	17	28	27	20	30	8	10	5	6	n/a	n/a
Infection		3	5	n/a	n/a	3	1	2	0	n/a	n/a	n/a	n/a
NICU admission		n/a	n/a	n/a	n/a	n/a	n/a	n/a	n/a	29	40	40	30
Hospitalization duration (days)		n/a	n/a	2.99 ± 1.26	2.95 ± 0.86	9.91 ± 3.06	11.17 ± 3.73	54 h	44 h	7.7 ± 6.3	9.1 ± 11	n/a	n/a

ETS: endotracheal suction; Non-ETS: non-endotracheal suction; MAS: meconium aspiration syndrome; n/a: data not available.

## Data Availability

Not applicable.

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
