# Peer review of "Routine Tracheal Intubation and Meconium Suctioning in Non-Vigorous Neonates with Meconium-Stained Amniotic Fluid: A Systematic Review and Meta-Analysis"

_diagnostics, 2022, doi:10.3390/diagnostics12040881_

Round 1
Reviewer 1 Report
Dikou and colleagues submit a systematic review and meta-analysis on routine tracheal intubation to suction meconium from non-vigorous neonates with meconium-stained amniotic fluid. The inclusion criteria for this systematic review are relatively loose, in that observational studies are accepted. The authors evaluate multiple outcomes and conclude that there is insufficient evidence to support routine tracheal intubation of neonates under these circumstances, which is in agreement with current recommendations from major international organizations.
More detailed comments:
In the introduction, Line 45, the authors state that MAS is eventually characterized by microbial pneumonia, but they provide no evidence to support or refute this statement. This is not a well-known fact, and indeed it may not be true. Actually, a Cochrane review suggests that antibiotics are not beneficial in MAS.
In the methods, the authors excluded studies published before 2012, but they did not provide a rationale for this methodological detail. According to the flow chart shown in figure 1, this resulted in 2 studies being excluded on the basis of date of publication. The authors should include the rationale for the exclusion of earlier dates, and possibly add to the discussion their evaluation of the conclusions if studies prior to 2012 had not been excluded.
Also in the Methods, the authors state that when the I2 statistic was greater than 50%, they used a random effects model for analysis. However, in the presentation of the Results, they sometimes used a fixed effects model even when heterogeneity was greater than 50%; this should be clarified.
Under Results, line 410, the word "ratio" is incorrect. The authors probably intended to write "difference".
In the Discussion, under Limitations, the authors refer to the heterogeneity of the studies included in their systematic review. However, they could further discuss the advantages and disadvantages of such heterogeneity. In particular, did the authors perform any sensitivity analysis in which they may have excluded observational studies? Also, the authors do not report on their grading of the evidence provided by each study. Did they perform any evaluation of the quality of the evidence in each study?
Minor issues:
There are several terms throughout the paper that may benefit from further English language editing, for clarity, but this is not critical for scientific accuracy. However, in line 129, the word "appearance" sounds awkward; perhaps it could be omitted altogether, or replaced by "development". Also, in various places in the discussion, (e.g., lines 439, 442) the word "aspiration" is used in a context where "suction" would be clearer; although not strictly incorrect, this would avoid confusion with the spontaneous aspiration of tracheal contents by the baby, which occurs in the opposite direction.
Finally, as a formatting issue, the titles of the articles listed in the references should use a consistent case.
Reviewer 2 Report
I have read this paper with great interest, and value the effort and the topic
However, there is a priority issue, as similar analyses on the very same topic (tracheal suction in non-vigorous neonates with meconium stained amniotic fluid) (Nangia et al, Cochrane Database Syst Rev 2021; and Phattraprayoon et al, Arch Dis Child Fetal Neonatal Ed 2021; Trevisanuto et al, Resuscitation 2020, and – be it not limited to non-vigorous cases, Wei et al, Am J Perinatol 2022.
It seems that this systematic review has not been registered, as otherwise, the authors would have noticed that other groups were already conducting as systematic review and meta-analysis. Otherwise, it seems that the PRISMA guidelines were well respected)
I therefore would like to invite the authors to compare their data to the other recent meta analysis, and try to put their work into perspective.
You suggest more RCTs, but based on the current dataset, wat would be in your opinion the primary outcome of interest, and related to this, how many cases are needed (power).
Author Response
REVIEWER 2
Dear reviewer,
We appreciate the time you took to review our paper and for your comments and contribution to the improvement of the paper before its potential publication. Your original comments are highlighted in italics in this document along with our response.
Reviewer comment: I have read this paper with great interest, and value the effort and the topic
However, there is a priority issue, as similar analyses on the very same topic (tracheal suction in non-vigorous neonates with meconium stained amniotic fluid) (Nangia et al, Cochrane Database Syst Rev 2021; and Phattraprayoon et al, Arch Dis Child Fetal Neonatal Ed 2021; Trevisanuto et al, Resuscitation 2020, and – be it not limited to non-vigorous cases, Wei et al, Am J Perinatol 2022.
It seems that this systematic review has not been registered, as otherwise, the authors would have noticed that other groups were already conducting as systematic review and meta-analysis. Otherwise, it seems that the PRISMA guidelines were well respected)
I therefore would like to invite the authors to compare their data to the other recent meta analysis, and try to put their work into perspective.
You suggest more RCTs, but based on the current dataset, wat would be in your opinion the primary outcome of interest, and related to this, how many cases are needed (power).
Thank you for your valid comment. We recognise that there is a priority issue as other meta-analyses have already been published on the very same topic; we mention two of them namely the ones of Phattraprayoon et al, Arch Dis Child Fetal Neonatal Ed 2021 and Trevisanuto et al, Resuscitation 2020. The third of Nangia et al, Cochrane Database Syst Rev 2021 has been added and discussed in the discussion section of the manuscript.
Nonetheless, the fact that many systematic reviews including ours have been conducted on the same topic, highlights the controversy over the resuscitation of the non-vigorous neonate manifested both in 2015 and 2021 guidelines revision. This is the reason that in our systematic review we chose to include the observational study of Oommen et al, Arch Dis Child Fetal Neonatal Ed 2021 that was published after the last guidelines publication and was not included in all previous meta-analyses. This study increased the sample size for 214 neonates.
Of course, we acknowledge that clinical trials are by definition of superior quality and preferable as primary studies to be included in a systematic review, although there are limitations such as ethical concerns and difficulty in implementation (voluntary withdrawal of patients from the intervention group). In our study we did not perform a de novo quality appraisal of the primary studies. In addition, we should always take into account the heterogeneity between even well designed RCTs. This is why we suggest more RCTs primarily for the outcome of death and MAS from a clinical perspective. We are not able at this stage to suggest a specific number of cases, as a specific power analysis was not in the scope of our study; however it is something that will concern us in the near future.A relevant statement was added in the discussion.
Round 2
Reviewer 2 Report
no additional comments or concerns
Author Response
Dear Reviewer,
We appreciate your contribution to the improvement of the paper.
Best regards,
The authors team